# Hepatocytic expression of human sodium-taurocholate cotransporting polypeptide enables hepatitis B virus infection of macaques

Benjamin J. Burwitz[1], Jochen M. Wettengel[2], Martin A. Mück-Häusl[2], Marc Ringelhan [2,3], Chunkyu Ko[2], Marvin M. Festag[2], Katherine B. Hammond[1], Mina Northrup[1], Benjamin N. Bimber[4], Thomas Jacob[5], Jason S. Reed[1], Reed Norris[4], Byung Park[6], Sven Moller-Tank[7,8,9], Knud Esser[2], Justin M. Greene[1], Helen L. Wu[1], Shaheed Abdulhaqq[1], Gabriela Webb[1], William F. Sutton[4], Alex Klug[4], Tonya Swanson[4], Alfred W. Legasse[4], Tania Q. Vu[5,10], Aravind Asokan[7,8,9], Nancy L. Haigwood[4], Ulrike Protzer [2,11] & Jonah B. Sacha[1,4]

Hepatitis B virus (HBV) is a major global health concern, and the development of curative therapeutics is urgently needed. Such efforts are impeded by the lack of a physiologically relevant, pre-clinical animal model of HBV infection. Here, we report that expression of the HBV entry receptor, human sodium-taurocholate cotransporting polypeptide (hNTCP), on macaque primary hepatocytes facilitates HBV infection in vitro, where all replicative intermediates including covalently closed circular DNA (cccDNA) are present. Furthermore, viral vector-mediated expression of hNTCP on hepatocytes in vivo renders rhesus macaques permissive to HBV infection. These in vivo macaque HBV infections are characterized by longitudinal HBV DNA in serum, and detection of HBV DNA, RNA, and HBV core antigen (HBcAg) in hepatocytes. Together, these results show that expressing hNTCP on macaque hepatocytes renders them susceptible to HBV infection, thereby establishing a physiologically relevant model of HBV infection to study immune clearance and test therapeutic and curative approaches.

[1] Vaccine and Gene Therapy Institute, Oregon Health and Science University, Beaverton, OR 97006, USA. [2] Institute of Virology, Technical University of Munich/Helmholtz Zentrum Munich, Munich 81675, Germany. [3] Department of Internal Medicine II, Klinikum rechts der Isar Technical University of Munich, Munich 81675, Germany. [4] Oregon National Primate Research Center, Oregon Health and Science University, Beaverton, OR 97006, USA. [5] Department of Biomedical Engineering, Oregon Health and Science University, Portland, OR 97239, USA. [6] Public Health and Preventative Medicine, Oregon Health and Science University, Portland, OR 97239, USA. [7] Gene Therapy Center, The University of North Carolina at Chapel Hill, Chapel Hill, NC 27599, USA. [8] Department of Genetics, The University of North Carolina at Chapel Hill, Chapel Hill, NC 27514, USA. [9] Department of Biochemistry and Biophysics, The University of North Carolina at Chapel Hill, Chapel Hill, NC 27599, USA. [10] Center for Spatial Systems Bioscience, Oregon Health and Science University, Portland, OR 97201, USA. [11] German Center for Infection Research, Munich partner site, Munich 81675, Germany. Benjamin J. Burwitz, Jochen M. Wettengel, and Martin A. Mück-Häusl contributed equally to this work. Ulrike Protzer and Jonah B. Sacha jointly supervised this work. Correspondence and requests for materials should be addressed to B.J.B. (email: burwitz@ohsu.edu)

HBV continues to infect humans worldwide despite the availability of an effective prophylactic vaccine since 1981, and it is estimated that one third of the world's population has been infected[1]. While the majority of adults clear HBV infection, exposure of young children or individuals with immune deficiencies mainly results in chronic infection. Currently, more than 260 million chronic HBV carriers are at high risk of developing serious liver pathology including cirrhosis and hepatocellular carcinoma. Hepatocellular carcinoma is the second leading cause of death from cancer worldwide and approximately half of these diagnoses are due to chronic hepatitis B virus infections[2]. Anti-viral drug therapies that employ reverse transcriptase inhibitors can suppress HBV replication, normalize inflammation, and slow progression of liver fibrosis[3]. However, these treatments are not curative due to persistence of cccDNA in the nuclei of infected hepatocytes, and do not eliminate cancer risk. While recombinant interferon-α can cure about 15% of patients treated, it is rarely used due to its severe side effects.

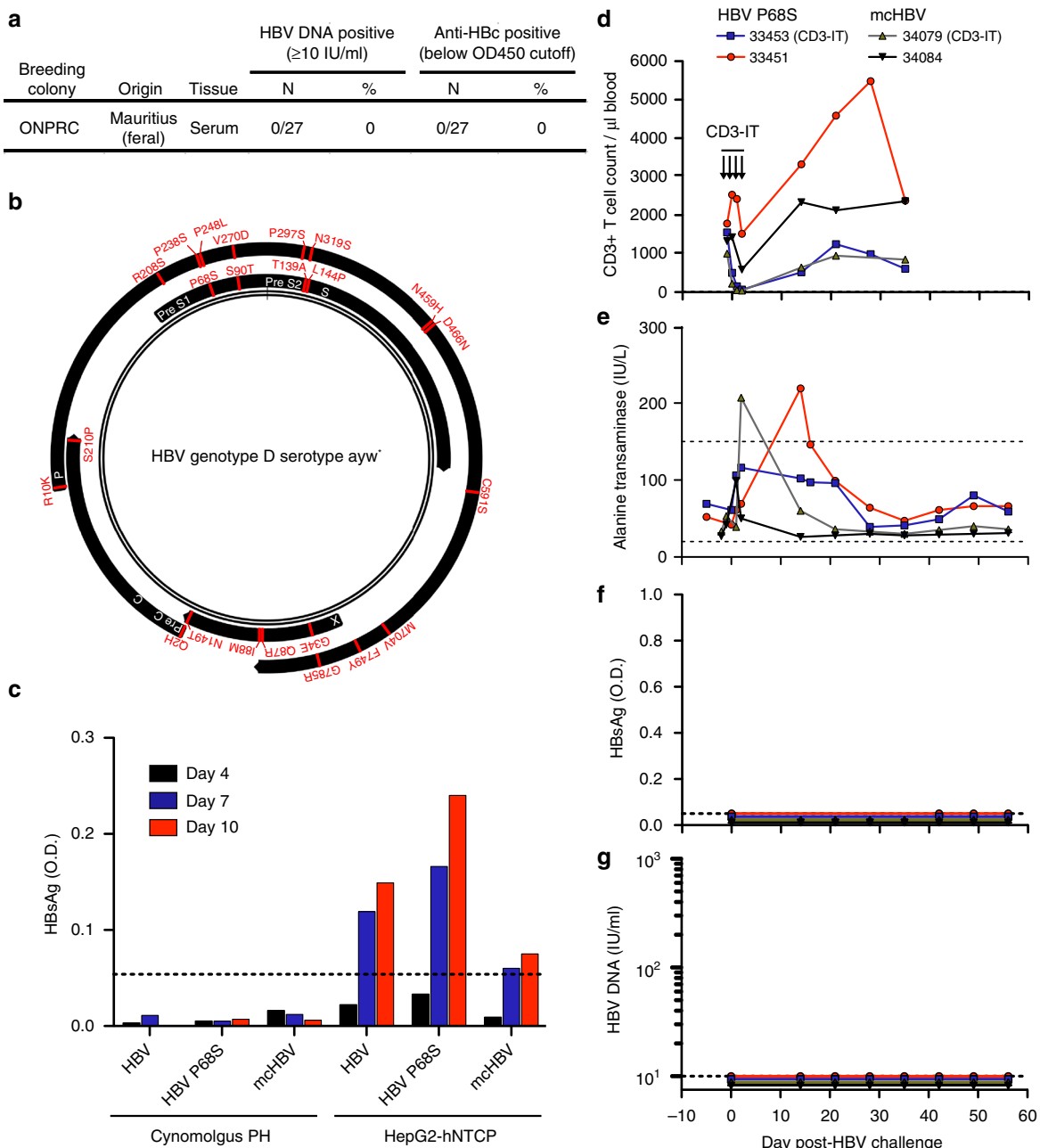

**Fig. 1** Mauritian-origin HBV (mcHBV) does not infect MCM in vitro or in vivo. **a** Twenty-seven MCM caught on the island of Mauritius and housed at the ONPRC were screened for HBV DNA and anti-HBc. **b** Schematic of HBV proteome (black) showing amino acid differences for the recently described mcHBV. * = HBV sequence described by Pasek et al.[18]. **c** Cynomolgus PH and HepG2-hNTCP cells were infected with HBV, HBV P68S, and mcHBV (MOI = 100) and infection monitored by HBsAg ELISA. Dotted lines represent limit of detection. **d**–**g** Four MCM were challenged with either HBV P68S or mcHBV (1 × 10^9 virions). Each condition represents a single-biological sample (N = 1). Figure is representative data of two separate experiments. **d** One MCM from each group received CD3-immunotoxin (CD3-IT) and absolute CD3+ T-cell frequencies were monitored over time. HBV infection was monitored longitudinally by measurements of **e** ALT (dotted lines represent normal ALT reference range in the ONPRC colony), **f** HBsAg (dotted lines represent limit of detection), and **g** HBV DNA (dotted lines represent limit of detection)

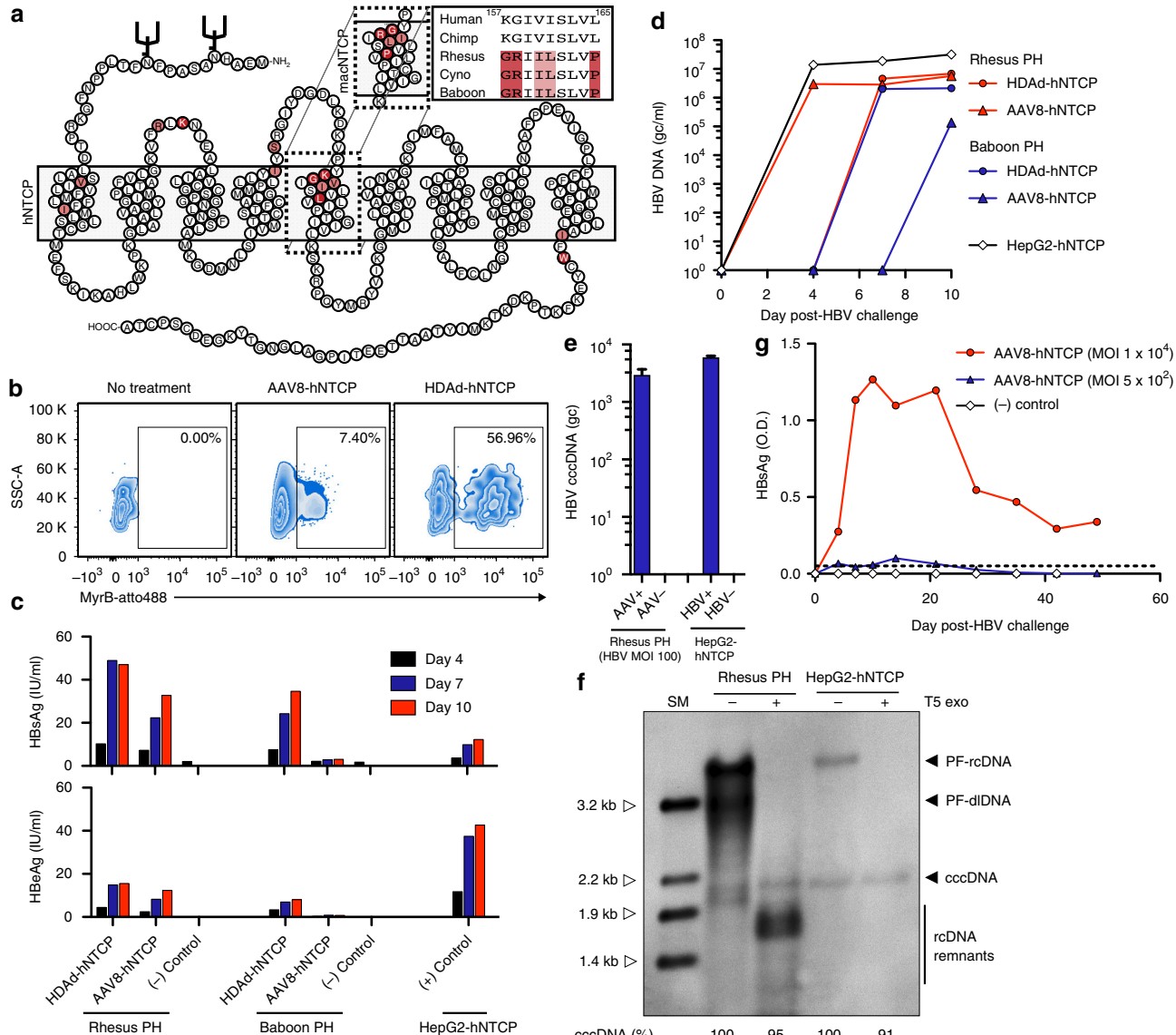

**Fig. 2** In vitro HBV infection of rhesus macaque PH. **a** Predicted schematic of NTCP showing amino acid differences between human and macaque NTCP. Differences in the sequences were labeled with lighter red for amino acid exchanges with similar physiochemical properties and darker red for exchanges with different physiochemical properties. Gray box represents cellular membrane. N-linked glycosylation sites represented by black brackets. macNTCP = macaque NTCP. **b** Rhesus macaque PH were transduced with either HDAd-hNTCP (MOI = 2) or AAV-hNTCP (MOI = $1 \times 10^4$) and stained 3 days later with Myrcludex B-atto488. **c** Rhesus macaque and baboon PH were transduced with either HDAd-hNTCP (MOI = 2) or AAV-hNTCP (MOI = $1 \times 10^4$) and infected with HBV (MOI = 100) 3 days later. Productive infection was monitored by quantification of HBsAg and HBeAg in the supernatant by ELISA. Each condition represents a single-biological sample (N = 1). Figure is representative data of two separate experiments. **d** HBV DNA qPCR on the same supernatants shown in **c**. Each condition represents a single-biological sample (N = 1). **e** Total intracellular DNA from $1 \times 10^6$ rhesus macaque PH and HepG2-hNTCP cells was used in a cccDNA-specific qPCR. Rhesus macaque PH transduced with AAV-hNTCP (MOI = $1 \times 10^4$) and infected with HBV (MOI = 100) 3 days later showed formation of cccDNA, while the non-transduced, HBV challenged PH did not. Bars represent standard error of measurement from two qPCR replicates. **f** Southern blot shows presence of cccDNA in rhesus macaque PH transduced with AAV-hNTCP (MOI = $1 \times 10^4$) and infected with HBV (MOI = 100). DNA was purified after Hirt extraction to remove protein-bound DNA forms. SM = size marker; T5 Exo = T5 exonuclease; PF-rcDNA = polymerase-free relaxed circular DNA; PF-dlDNA = polymerase-free duplex linear DNA. Figure is representative data of two separate experiments. **g** Neonate rhesus macaque PH were transduced with AAV-hNTCP (MOI = $1 \times 10^4$ or $5 \times 10^2$) and infected with HBV (MOI = 100) 3 days later. HBV infection was then monitored longitudinally by HBsAg ELISA. Each condition represents a single-biological sample (N = 1)

Thus, there remains an urgent need to better understand HBV clearance, and to identify and test innovative curative HBV therapies.

Multiple animal models of HBV infection have been utilized over the past four decades including the Peking duck[4, 5], woodchuck[6–8], mouse[9, 10], and chimpanzee[11–13]. These models have contributed significantly to our understanding of virus-host interactions, but each have drawbacks that limit their utility for

drug development. First, ducks and woodchucks are infected with species-specific hepatitis viruses that are related to HBV, but differ in genomic sequence, protein composition, and virion structure[14]. In addition, both animals are outbred, ill-defined genetically, and scarcely used outside the HBV field, which consequentially limits the availability of important reagents, such as major histocompatibility complex:peptide tetramers and immunophenotyping antibodies. Second, because HBV does not infect

mouse hepatocytes, current mouse models rely on artificial expression of HBV, either as a germline transgene or as a linear genome transferred by viral vectors[15], or on humanized livers that require significant immunomodulation to maintain. Additionally, murine hepatocytes do not support the formation of cccDNA, restricting their use in cure research. The relatively short lifespan of mice further detracts from the model, as liver dysfunction and fibrosis are generally gradual processes that can take years to manifest in humans. Finally, the NIH recently suspended all funding for chimpanzees in biomedical research, effectively ending the only physiologically relevant HBV animal model. Thus, development of a new non-human primate model where immune responses and cccDNA clearance can be studied using readily available reagents would mark a milestone in HBV research.

Dupinay et al.[16] recently described endogenous HBV circulating in a geographically isolated cynomolgus macaque population on the island of Mauritius (mcHBV). Mauritian cynomolgus macaques (MCM) descended from a founder population left on the island ~500 years ago, and due to the absence of endogenous HBV in non-Mauritian cynomolgus macaques, Dupinay et al.[16] conjectured that mcHBV was transmitted from humans to MCM since the time of their population bottleneck. Indeed, the mcHBV sequence is 99% identical to clinical isolates of HBV genotype D serotype ayw3.

Here we show that MCM housed at the Oregon National Primate Research Center (ONPRC) have no evidence of current or prior HBV infection in the blood, as measured by HBV DNA qPCR and anti-HBc ELISA, respectively. Subsequent high-dose challenge of MCM with recombinant mcHBV did not result in productive infection. Therefore, in a parallel attempt to overcome the HBV species-barrier, we transduced rhesus macaque hepatocytes with hNTCP, followed by infection with a recombinant, clinical isolate of HBV. We show that non-human primate hepatocytes can support HBV replication in vitro following transduction with viral vectors encoding hNTCP. Remarkably, when we administered these viral vectors to rhesus macaques and challenged with HBV, we detected increasing amounts of HBV DNA in the serum over time, indicating active HBV replication in vivo. Further analysis established replication in the liver, where we detected HBV DNA, RNA, and HBV core antigen (HBcAg) in hepatocytes. These findings pave the way for a physiologically relevant, rhesus macaque HBV infection model that can be utilized to better understand HBV clearance and for the testing of curative HBV treatments.

## Results

**mcHBV challenge of MCM.** Dupinay et al.[16] recently published the discovery of mcHBV, a naturally occurring HBV in MCM, whereby 25.8% of MCM serum samples tested positive for HBV DNA (range: $10^1$–$10^6$ copies per ml). In order to confirm this frequency of HBV infection, and to identify MCM with the highest viral loads for serum transfer experiments, we collected serum samples from 27 wild-caught MCM imported from Mauritius to ONPRC and performed a previously published, clinically validated HBV DNA qPCR assay[17]. In addition, we screened these serum samples for anti-HBc seroconversion using a clinical, competitive ELISA assay to identify any MCM with previously cleared HBV infections. We did not detect HBV DNA or anti-HBc seroconversion in any of the MCM serum samples tested at ONPRC (Fig. 1a, Supplementary Table 1). Given that the probability of discovering zero infected MCM out of 27 MCM is only 0.0317% using an exact binomial distribution based on the prevalence reported by Dupinay et al.[16], these results indicate that

mcHBV is likely not endogenous in MCM on the island of Mauritius.

The published sequence of mcHBV (GenBank accession#: HE815465.1) is 99% identical to clinical isolates of HBV genotype D serotype ayw3. There are a total of 23 amino acid differences between a commonly used HBV genotype D serotype ayw strain first described by Pasek et al.[18] and mcHBV, distributed throughout all four HBV open reading frames (Fig. 1b). To test the hypothesis that a single mutation within the Pre-S1 region, P68S, enables mcHBV binding to macaque NTCP and facilitates mcHBV entry and replication in MCM hepatocytes[16], we generated recombinant, high-titer stocks of mcHBV and HBV P68S (containing only the P68S mutation in Pre-S1) based on the HBV backbone described by Pasek et al.[18]. We confirmed the genomic sequences of these stocks by deep sequencing and infected cynomolgus macaque primary hepatocytes (PH) or HepG2 cells expressing hNTCP (HepG2-hNTCP) at a multiplicity of infection (MOI) of 100 virions (Supplementary Fig. 1). Another commonly used HBV genotype D serotype ayw clinical isolate, purified from HepAD38 cells and further referred to as HBV[19], was used as a positive control virus in the assay. We observed replication of all 3 viruses in HepG2-NTCP, but did not detect HBV replication in cynomolgus macaque PH nor secretion of HBV surface antigen (HBsAg) by ELISA (Fig. 1c).

Next, we challenged MCM in vivo with mcHBV and HBV P68S. We challenged four MCM intravenously with $1 \times 10^9$ virions (Dane particles) of either mcHBV ($N = 2$) or HBV P68S ($N = 2$) and monitored for signs of infection over the course of 8 weeks. In addition, we depleted CD3+ T cells in one MCM from each group by administration of anti-CD3-immunotoxin[20] to remove early T-cell immunity and provide the highest chance of seeding an acute HBV infection (Fig. 1d). We found elevated ALT activity in serum at single-time points from two MCM during the first few weeks post-challenge, and animal 33451 exhibited increases in absolute CD3+ T cell counts over the first 4 weeks post-infection (Fig. 1d, e). However, these changes were not associated with markers of productive HBV infection including serum HBsAg ELISA, HBV DNA qPCR, and anti-HBc ELISA (Fig. 1f, g, Supplementary Table 2). Cumulatively, these results cast doubt on the use of mcHBV in MCM as a robust and reproducible pre-clinical model of HBV, and suggest that alternate approaches are necessary to generate a non-human primate model of HBV.

**HBV replicates in non-human primate PH expressing hNTCP.** HBV requires hNTCP to enter hepatocytes, and the HBV binding region of hNTCP has been identified (amino acids 157–165)[21]. Macaque NTCP harbors 5 amino acid differences within this region, precluding its binding to the pre-S1 region of the large HBV envelop protein (Fig. 2a)[21–23]. We hypothesized that HBV would replicate efficiently in non-human primate PH following binding and entry. To test this hypothesis, we generated helper-dependent adenovirus (HDAd, serotype 5) and adeno-associated virus (AAV, serotype 8) vectors expressing hNTCP and transduced freshly isolated rhesus macaque and baboon PH in vitro. Three days after HDAd-hNTCP or AAV8-hNTCP transduction we detected expression of hNTCP on the surface of PH by surface staining with the hNTCP binding, pre-S1-derived peptide Myrcludex B (Fig. 2b). Following confirmation of hNTCP surface expression, we challenged parallel transduced PH with HBV at an MOI of 100 virions per cell.

Rhesus macaque and baboon PH expressing hNTCP supported high levels of HBV infection, evidenced by increasing supernatant concentrations of both HBsAg and HBeAg, whereas non-transduced PH did not (Fig. 2c). In addition, supernatant qPCR

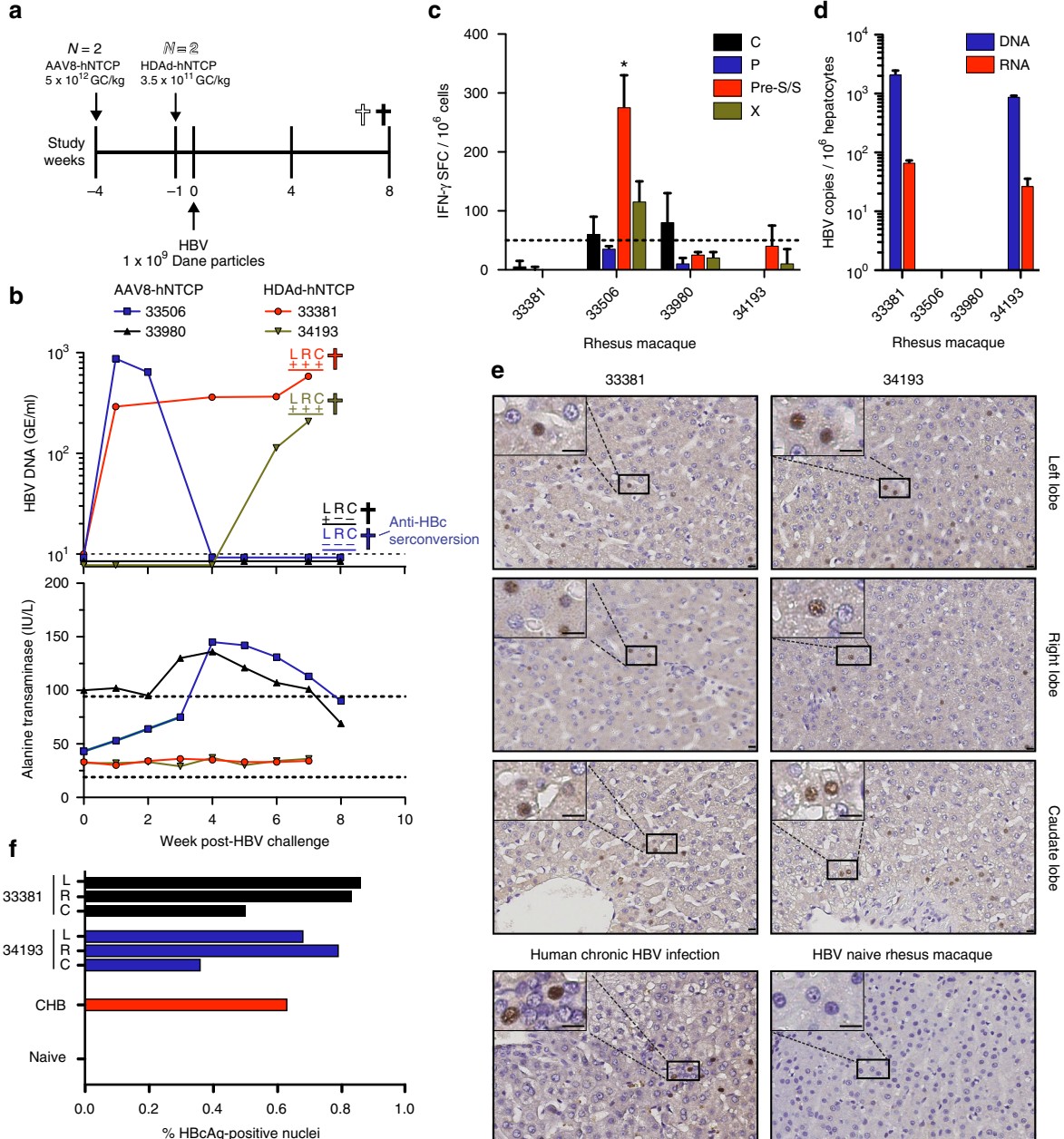

**Fig. 3** In vivo HBV infection of rhesus macaques. **a** Timeline showing administration of viral vectors encoding hNTCP, HBV challenge, and necropsy of rhesus macaques. GC = genome copies. **b** Longitudinal monitoring of HBV infection by HBV DNA (dotted line represents assay limit of detection) and alanine transaminase concentrations (dotted lines represent normal alanine transaminase reference range in ONPRC colony) in the serum. **c** IFN-g ELISpot measurements of liver-resident T-cell responses against HBV C, P, S, and X. * = Response met statistical significance, defined as mean number of spot forming cells (SFCs) of triplicate sample wells exceeding background (no stim) plus two standard deviations in a homoscedastic t-test. Dotted line represents assay limit of detection. **d** Levels of HBV DNA and RNA as assessed by qPCR and qRT-PCR, respectively, in rhesus macaque PH isolated from the liver via collagenase media perfusion. **e** HBcAg immunohistochemistry from HBV-infected rhesus macaque liver lobes, HBV naive rhesus macaque liver, and chronically HBV-infected human liver. Size bars indicate distance of 10 μm. **f** Quantification of HBcAg immunohistochemistry in **e** showing average frequency of HBcAg-positive nuclei (include parenchymal and non-parenchymal cells) from indicated liver lobes. Human CHB and naive rhesus macaque samples represent a single-biological sample (N = 1). CHB = Human chronic HBV infection, L = left liver lobe, R = right liver lobe, C = caudate liver lobe

revealed packaging of HBV DNA, indicating the formation and release of Dane particles (Fig. 2d). Indeed, we detected both Dane particles (~40 nm diameter) and spherical bodies (~15 nm diameter) in supernatants collected from HBV-infected rhesus macaque and baboon PH (Supplementary Fig. 2). These spherical bodies represent subviral HBV particles and were found at a

much higher frequency than Dane particles within the supernatant. Finally, and most importantly, we detected the formation of T5 exonuclease-resistant HBV cccDNA in rhesus macaque PH by qPCR and Southern blotting (Fig. 2e, f). T5 exonuclease readily digested relaxed-circular (rcDNA) and duplex-linear DNA (dlDNA), leading to visualization of DNA remnants, a

phenomenon previously described[24]. Thus, rhesus macaque and baboon hepatocytes expressing hNTCP support the entire HBV replication cycle.

To further confirm these findings and to assess the durability of HBV infection on rhesus macaque PH, we transduced rhesus macaque PH with a high ($1 \times 10^4$) or low ($5 \times 10^2$ genome copies) dose of AAV8-hNTCP and monitored HBV infection longitudinally (Fig. 2f). We found that HBsAg levels in the supernatant were directly related to the transducing AAV8-hNTCP dose and that HBV infection could be sustained on rhesus macaque PH in vitro for as long as 49 days (Fig. 2g). Thus, expression of hNTCP on rhesus macaque PH renders these cells permissive to HBV infection for prolonged periods of time, indicating that chronic HBV infection should be possible in rhesus macaques.

**HBV infects rhesus macaques expressing hNTCP in vivo.** Given our in vitro results demonstrating that hNTCP-expressing macaque PH can support HBV infection for extended periods of time, we next tested whether hNTCP-expression on hepatocytes in the liver could facilitate HBV infection in vivo. We injected four rhesus macaques with either HDAd-hNTCP ($N = 2$) or AAV8-hNTCP ($N = 2$) and subsequently challenged these animals with HBV (Fig. 3a). Importantly, we confirmed liver transduction and hNTCP expression by qPCR and qRT-PCR, respectively, showing hNTCP expression in all tested lobes (Supplementary Fig. 3). We also observed no cellular or humoral immunity against hNTCP in these animals (Supplementary Fig. 4). We sent blinded serum samples to the Oregon Health and Science University clinical diagnostic laboratory for detection of HBV DNA and detected HBV viral loads in several consecutive blood samples from three animals, showing that HBV is capable of infecting and replicating in macaque livers in vivo following transduction with hNTCP expressing vectors (Fig. 3b–top panel). Importantly, as we first detected HBV DNA in the serum of animal 34193 at week 6 post-challenge, we can exclude the possibility that we are simply detecting input virus. In further support of this, we were unable to detect HBV P68S and mcHBV in MCM after inoculation with identical virus doses. We also monitored changes in serum ALT levels following HBV challenge (Fig. 3b–lower panel). Rhesus macaque 33506 showed elevated levels of serum ALT, concomitant with a drop in HBV viral load to undetectable levels. This temporal relationship between elevated serum ALT and clearance of circulating HBV DNA raised the potential of immune-mediated clearance of HBV infection. Consistent with HBV DNA and ALT results obtained from animal 33506, IFN-γ ELISpot analysis using liver-resident lymphocytes from this animal showed a statistically significant T cell response against a preS/S peptide pool (Fig. 3c). Less pronounced responses against HBV C and X were also detected (Fig. 3c). We also assessed anti-HBc seroconversion in all four rhesus macaques at the final time point and found anti-HBc seroconversion in 33506, but in none of the other animals (Fig. 3b–top panel, Supplementary Table 3). Finally, we measured commonly assessed serological markers of HBV infection, including anti-HBs, anti-HBe, and HBsAg at the final time point, and found none of these markers, suggesting that increased hNTCP expression across macaque hepatocytes will be necessary to observe these parameters (Supplementary Table 4).

Next, we examined the liver for the presence of HBV replication. We extracted total DNA from liver tissue taken from the left, right, and caudate lobes and detected HBV DNA in three HBV challenged animals by nested PCR, showing that HBV replication is occurring in liver tissue (Fig. 3b–top panel and Supplementary Fig. 5).

Because it was possible that the HBV DNA detected could be from blood perfusing the liver sections, we also obtained highly-purified hepatocytes from the right liver lobes of all four rhesus macaques by collagenase perfusion. We performed qPCR and qRT-PCR for the detection of HBV DNA or RNA, respectively, in this population of purified hepatocytes. Supporting our nested PCR data, we found both HBV DNA and RNA in hepatocytes of animals 33381 and 34193, but not in hepatocytes from 33506 and 33980 (Fig. 3d). Thus, analysis of HBV DNA and RNA levels in the liver of 33506 further support our assertion of immune-mediated HBV clearance in this animal.

Finally, we performed IHC on liver sections collected from the left, right, and caudate lobes of rhesus macaques 33381 and 34193 to determine the level of HBcAg expression. We compared these results to the livers of a chronically HBV-infected human and rhesus macaque 34200, which was treated with AAV8-hNTCP but never HBV challenged. Importantly, we found HBcAg staining in all sections analyzed from HBV-infected rhesus macaques (Fig. 3e). HBcAg staining was localized to small pockets of expression with only 0.5–1.0% of nuclei in HBV-infected rhesus macaque livers staining HBcAg-positive (includes parenchymal and non-parenchymal cells), supporting the low-level HBV replication observed in the serum (Fig. 3f). This frequency of HBcAg-positive nuclei was comparable to the liver section acquired from a chronically HBV-infected human (Fig. 3f). Taken together, our data show for the first time that HBV infects rhesus macaques expressing hNTCP and that viral replication can be sustained for at least 6 weeks in vivo, leading to the induction of HBV-specific T cell responses.

## Discussion

Our study presents evidence that experimental HBV infection of rhesus macaques is possible, paving the way for testing of novel HBV therapies in a well-characterized non-human primate infectious disease model. We show that after introducing hNTCP, HBV enters and replicates in the livers of rhesus macaques. This replication was associated with HBV-specific cellular and humoral immunity in the animal with the highest HBV viral load, indicating that higher HBV replication may be required to consistently induce these immune responses. This is an important finding, as one advantage of the rhesus macaque model over other current HBV models is the closely related immune systems shared between humans and macaques[25–27]. Indeed, many of the emerging HBV therapies rely on modulating anti-HBV immunity, rather than targeting the virus directly[9, 28, 29]. Thus, we believe that further characterization of this model will provide an important resource to study immune control of HBV and for testing HBV therapies targeting host immunity.

We were unable to recapitulate the finding that a particular strain of HBV reported earlier infects MCM using both in vitro and in vivo studies[16]. The reasons for this remain unclear, but our findings indicate that the zoonotic transmission of HBV from human to MCM may have occurred within the particular colony studied and not on the island of Mauritius. Alternatively, there may be hurdles to overcome with using recombinantly produced, high-titer HBV stocks rather than the simple transfer of serum between HBV-infected animals. Indeed, the majority of animal HBV research relies on transmission of HBV through serum transfer. We were, however, also unable to detect any HBV related virus in any of the captured animals tested. Regardless of our inability to repeat the results of Dupinay et al.[16], we show here that expressing hNTCP on the surface of macaque hepatocytes is sufficient for HBV entry and replication.

It is well established that HBV does not bind to macaque NTCP. Indeed, Schieck et al.[23] have elegantly shown that

radiolabeled peptides corresponding to the NTCP binding region of pre-S1 are retained in the livers of mice, rats, and dogs, but not cynomolgus macaques[21, 23]. However, the binding of pre-S1 peptides (and by extension HBV particles) to the livers of mice, rats, and dogs does not render these species susceptible to productive HBV infection, indicating a post-entry host-restriction preventing productive HBV infection. Importantly, we show here that no post-entry restriction of HBV infection exists in rhesus macaques, as replication of HBV in rhesus macaque PH was abundant following infection. Thus, expression of hNTCP is the only requirement for infection of macaque hepatocytes in vitro and in vivo.

We also found that cccDNA is formed in HBV-infected rhesus macaque PH. This is an important difference to current mouse models in which cccDNA is not formed following HBV infection, precluding their use in HBV cure research[30, 31]. In addition, because all steps of the HBV life cycle are supported by rhesus macaque PH, in vitro findings can be rapidly transitioned within the same species to an in vivo model. This stands in contrast to in vitro findings using HBV-infected human PH, which require immunomodulated humanized mice to transition to the in vivo setting. While we were unable to detect cccDNA in hepatocytes in vivo, this may be due to the low level of HBV replication. Indeed, our in vitro work with rhesus macaque PH shows that cccDNA is formed, indicating that further assessment of cccDNA formation in rhesus macaques in vivo is necessary.

One hurdle currently facing this macaque HBV model is the requirement to transduce high numbers of hepatocytes in vivo with hNTCP expressing viral vectors. In our study, we utilized HDAd and AAV vectors to deliver hNTCP to the liver. To achieve efficient hepatocyte targeting, we used an AAV serotype 8 capsid, and hNTCP was expressed under control of a ubiquitous chicken beta-actin promoter. In animals transduced with AAV8-hNTCP, HBV infection was variable, with only one animal exhibiting HBV DNA in serum following challenge. Our HDAd vector expressed hNTCP under the control of a hepatocyte specific transthyretin promoter, and both animals exhibited active viral replication as assessed by serum HBV DNA quantification. Unfortunately, we were unable to determine the number of macaque hepatocytes expressing hNTCP due to cross-reactivity of all available antibodies with macaque NTCP, and loss of the Myrcludex B epitope on hNTCP during formalin fixation. However, the limited number of hNTCP DNA copies per diploid genome detected in our qPCR assay indicates that hepatocyte transduction was low. Indeed, if compared to previous AAV transduction studies in mouse and rhesus macaque livers, the number of hNTCP DNA copies per diploid genome we detected corresponds to <5% of total hepatocytes transduced[32, 33]. Superior tools for the delivery of hNTCP to a greater number of hepatocytes, or experiments looking at persistent transduction or integration of hNTCP into neonate hepatocytes may yield better hNTCP expression and thus higher infection levels. This particular concept is additionally appealing in that HBV infection of neonates results in the highest frequency of chronic HBV infection.

The finding that hNTCP expression on non-human hepatocytes facilitates HBV infection is not unprecedented. Lempp et al.[34] recently described HBV infection of AAV-hNTCP transduced pig and macaque PH in vitro. The range of hosts susceptible to HBV infection following hNTCP-mediated viral entry is unknown. However, we found that expression of hNTCP on the surface of baboon PH was also sufficient to allow productive infection, indicating that the ability of HBV to replicate post-entry may pertain to a wide range of non-human primate species. Indeed, we believe further research into alternative non-human primate species is warranted.

Given the urgent need for curative HBV treatments, we believe our macaque model is emerging at a pivotal time and will facilitate critical translational research. We show that HBV replicates in rhesus macaques in vivo, forms cccDNA in PH, and induces anti-viral cellular and humoral immunity. Thus, the main elements of human HBV infection can be recapitulated in macaques. This macaque HBV model may be used to ensure physiologically relevant, pre-clinical testing of emerging therapies.

## Methods

**Experimental design.** Given the exploratory nature and the binary readout (HBV infection vs. no HBV infection) of this study, we utilized small animal groups. Randomization was not used to assign animals to their respective experimental groups, and authors were not blinded to assignments. A total of 4 Mauritian cynomolgus macaques (33451-F-4 yr, 33453-F-5 yr, 34079-F-5 yr, and 34084-M-6 yr) and 4 Indian rhesus macaques (33381-F-1 yr, 33506-F-1 yr, 33980-F-2 yr, and 34193-F-1 yr) were challenged intravenously with $1 \times 10^9$ Dane particles of HBV and followed longitudinally in this study. Anti-CD3-immunotoxin (NHP AIDS Reagent Resource) was given intravenously twice daily at 0.025 mg per kg. AAV8-hNTCP ($5 \times 10^{12}$ genomic copies per kg) and HDAd-hNTCP ($3.5 \times 10^{11}$ genomic copies per kg) were administered intravenously at day −28 and day −7 post-HBV infection, respectively. Necropsy dates were chosen to ensure active HBV replication in the liver at the time of killing. Animals were cared for at the Oregon National Primate Research Center (ONPRC) with the approval of the ONPRC Animal Care and Use Committee using the standards of the NIH Guide for the Care and Use of Laboratory Animals.

In vitro and ex vivo experiments were performed in duplicate where possible. When duplicate samples were not attainable, additional experiments were performed to ensure repeatable results. For example, we have now shown in vitro HBV infection of rhesus macaque PH in at least three separate experiments. All data collected are shown, and no outliers were excluded.

We sent blinded serum samples to the Oregon Health are Science University clinical diagnostic laboratory for detection of HBV DNA in vivo.

**Production of recombinant HDAd.** The bacterial artificial chromosome (BAC) containing the genome of the helper-dependent adenovirus HDAd-hNTCP was constructed by serial cloning steps based on homologous recombination as described previously[35]. In brief, an expression cassette containing the human Na+-taurocholate cotransporting polypeptide (hNTCP—GenBank Accession #JQ814895.1) under control of the liver-specific transthyretin (TTR) promoter derived from the plasmid pCH-TTR-GFP[36] and the human beta globin poly-adenylation sequence derived from the plasmid pRTS1[37] was cloned into the 3′ region of a helper-dependent adenoviral vector genome encoded by the bacterial artificial chromosome BAC-HCAdV5-CMV/eGFP[35].

The respective helper-dependent adenoviral vector HDAd-hNTCP was produced as described previously[38]. In brief, 116 producer cells (provided by Dr. Philip Ng), which are HEK293 cells constitutively expressing the Cre recombinase[39], were transfected with MssI-digested BAC containing the HDAd-hNTCP genome and infected with helper-virus AdNG163R-2 (MOI 3) (provided by Dr. Philip Ng)[39]. Initially generated vector particles collected 48 h post-infection were used for five sequential amplification steps utilizing increasing numbers of 116 cells. For large-scale amplification, 116 producer cells were cultivated in suspension in a 3 L bioreactor. Collected vector particles were purified using two consecutive CsCl-gradients and subsequently dialyzed against a physiological buffer (10 mM Tris (pH 8.0), 2 mM MgCl2, 4 % sucrose (w/v)).

**Production of recombinant AAV.** Recombinant AAV8-hNTCP vectors were generated by triple-plasmid transfection into HEK293 cells (ATCC Cat# CRL-1573). Briefly, HEK293 cells were transfected with pXR8 (provided by the UNC vector core), encoding AAV replication and capsid genes, pXX6-80 (provided by the UNC vector core), encoding adenoviral helper genes, and pTR-hNTCP (plasmid pTR provided by the UNC vector core), encoding AAV2 inverted terminal repeats (ITRs) flanking a transgene cassette expressing human NTCP as driven by the chicken beta-actin promoter. Cells were maintained in DMEM (Gibco) supplemented with 5% fetal bovine serum (FBS) (Sigma-Aldrich) and 1% penicillin/streptomycin. Supernatant was collected from cells 72 and 144 h post-transfection, pooled, and concentrated by PEG precipitation. Virus was purified by iodixanol gradient ultracentrifugation and Zeba Spin desalting column centrifugation (Thermo Scientific) and vector titers determined by quantitative PCR using a Roche Lightcycler 480.

**Generation of HBV stocks.** A high-titer virus stock of HBV was produced by culturing HBV producing HepAD38 cells (provided by Dr. Chris Seeger) and collecting the HBV-containing supernatant every 3–4 days. Purification of HBV was performed via heparin columns and sucrose gradient ultracentrifugation as previously described[40]. HBV stocks were titered by HBV DNA qPCR of enveloped, heparin-binding DNA-containing particles.

P68S was introduced by site-directed mutagenesis into the plasmid pT-HBV1.3 (GenBank Accession # V01460)[18] containing the HBV 1.3-fold genotype D, subtype ayw3 genome. For generation of mcHBV, we synthesized a mcHBV 1.3-fold genome as previously described by Dupinay et al.[16] and inserted into the same plasmid to form pT-mcHBV1.3. HepG2 cells (ATCC Cat# HB-8065) were transfected with pT-HBV1.3-P68S and pT-mcHBV1.3 and selected for 3 weeks. Single-cell clones were finally selected by high-level secretion of HBsAg (determined by ELISA) and HBV DNA (by qPCR) in cell culture supernatants. Concentration of HBV P68S and mcHBV was performed via heparin column purification and subsequent sucrose gradient ultracentrifugation as previously described[40].

HBV DNA was extracted from 20 µl of each HBV stock using the QiaAmp Minelute Virus Spin kit (Qiagen) according to manufacturer's instructions. HBV DNA was amplified by PCR using two overlapping amplicons with the following primer sets: (1) HBV_Illumina-F1 (5′-CTCTCGTTTTTGCCTTCTGAC-3′) and HBV_Illumina-R1 (5′-GCCTCCTAGTACAAAGACCTTTAA-3′), (2) HBV_Illumina-F2 (5′-ATCGGGACTGATAACTCTGTTG-3′) and HBV_Illumina-R2 (5′-TCCCCACCTTATGAGTCCAA-3′). Next, the overlapping PCR amplicons were gel purified using in 1.5% agarose gel followed by cleanup with the Nucleospin Gel and PCR Cleanup Kit (Macherey-Nagel, Bethlehem, PA), following the manufacturer's instructions. Purified DNA was quantified using a Qubit (Invitrogren, Carlsbad, CA), and the amplicons were pooled at equimolar amounts. These pools were fragmented and barcoded using the Nextera XT Kit (Illumina, San Diego, CA), followed by purification using Ampure XP beads (Beckman Coulter, Brea, CA). The resulting libraries were sequenced using an Illumina MiSeq (Illumina, San Diego, CA). Sequence data were processed using a custom analysis pipeline written by B.N.B. This pipeline has been made available through DISCVR-Seq (https://github.com/bbimber/discvr-seq), as a module for LabKey Server, an open-source platform for the management of scientific data[41]. Briefly, raw reads were trimmed by sequence quality using Trimmomatic[42] and aligned against the Galibert et al. HBV reference using the aligner BWA-Mem[43]. Local re-alignment around indels was performed using GATK[44]. Nucleotide difference between reads and the reference sequences were scored using custom software that utilized HTS-JDK (http://samtools.github.io/htsjdk/), and amino acid translations were performed that accounted for read-specific flanking sequence.

**Isolation of macaque PH**. A single lobe of non-human primate liver was perfused with 250 ml of Hanks Balanced Salt Solution (HyClone) followed by perfusion with 100 ml collagenase media (DMEM/F12 (Gibco), 3% bovine growth serum (HyClone), supplemented with L-glutamine (HyClone), antibiotic/antimycotic (HyClone), Gentamicin (Life Technologies)), and to removed blood from the tissue. This was followed by re-circulation of 150 ml collagenase media through the liver lobe at 42 °C for 1 h. Following collagenase perfusion, the liver was filleted with scalpels and then physically separated with a 3 ml syringe plunger. Cells were first filtered through a tea strainer to remove large debris, followed by a 70 µM filter to ensure single-cell suspension. Cells were then washed three times in PH media (DMEM/F12 (Gibco), 10% bovine growth serum (HyClone), 23 mM HEPES buffer (HyClone), 0.6 mg/ml glucose, supplemented with L-glutamine (HyClone), antibiotic/antimycotic (HyClone), and Gentamicin (Life Technologies)) at 4 °C. Centrifugation for each wash was: (1) 100×g for 3 min, (2) 70×g for 3 min, and (3) 50×g for 3 min, all at 4 °C. Purified hepatocytes were then resuspended in 37 °C PH media and counted. Hepatocytes were plated onto collagenized 12-well plates at $7.5 \times 10^5$ per well and allowed to adhere overnight. The next day, wells were washed three times with HBSS and cells cultured in 1 ml PH media supplemented with 1.8% DMSO (PH-DMSO).

**Transduction and HBV infection of macaque PH**. Following 24 h of culture in PH-DMSO, HDAd and AAV8 vectors expressing hNTCP were added to the culture for 3 days. Hepatocytes were then washed twice in 5 ml HBSS and HBV-containing media (PH-DMSO containing 4% PEG6000) was added overnight. The next morning, wells were washed five times with 5 ml HBSS and then cultured in 1 ml PH-DMSO for the remainder of the experiment.

**HBV DNA and RNA quantification**. To quantify HBV in supernatant or serum, total DNA was extracted from 200 µl using the QiaAmp Minelute Virus Spin kit (Qiagen) according to manufacturer's instructions. For analysis of cell associated HBV, total intracellular DNA and RNA were extracted using the AllPrep DNA/RNA kit according to manufacturer's instructions (Qiagen).

Total HBV DNA and RNA were quantified using the TaqMan Fast Advanced Master Mix (Applied Biosystems) and AgPath-ID One-Step RT-PCR kit (Life Technologies), respectively. Total HBV DNA/RNA primers and probe were: HBV_qPCR-F (5′-GGCCATCAGCGCGTGA-3′), HBV_qPCR-R (5′-TGCTGCGA GCAAAACA-3′), and HBV_qPCR-Probe (5′-6FAM-CTCTGCCGATCCATACTG CGGAACTC-TAMRA-3′) using an annealing temperature of 60 °C. All thermocyling parameters followed exactly to suggested manufacturers instructions. All thermocycling and quantification measurements were conducted on an Applied Biosystems Step One PLUS. Quantification was assessed relative to an absolute standard curve using the plasmid pCEP4 with the targeted insert as template.

For cccDNA quantification, intracellular total DNA was extracted using the NucleoSpin tissue kit (Macherey-Nagel) after lysing cells in one well in 200 µl buffer (12-well plate) according to manufacturer's instructions. DNA was eluted from NucleoSpin Tissue Columns with 100 µl pre-warmed (70 °C) buffer and subjected to T5 exonuclease treatment (New England Biolabs) at 37 °C for 30 min. The mixture was diluted eightfold before using as template in the qPCR assay.

cccDNA was quantified using a previously described primer/probe set spanning the nick and gap in the HBV genome, preferentially amplifying the completed cccDNA minichromosome over the rcDNA in viral capsids. A second primer/probe set targeted the S gene amplifying both rcDNA and cccDNA using TaqMan Fast Advanced Master Mix (Applied Biosystems). Primer/probe sets were: (1) HBV_CCCF2 (5′-CCGTGTGCACTTCGCTTCA-3′), HBV_CCCR2 (5′-GCACA GCTTGGAGGCTTGA-3′), and HBV-CCCP2 (5′-6FAM- CATGGAGACCAC CGTGAACGCCC–BHQ1-3′), and (2) HBV_S-F (5′-TGGCCAAAATTCGCAGT CCC-3′), HBV_S-R (5′-AGATGAGGCATAGCAGCAGGAT-3′), and HBV_S-P (5′-6FAM-ATGATAAAACGCCGCAGACACATCCAGC–BHQ1-3′). using an annealing temperature of 60 °C. Quantification was assessed separately for each primer/probe target set against an absolute standard curve using the plasmid pcDNA_HBV1.3. To control for low-level amplification of rcDNA with the primer/probe set targeting the nick and gap of the HBV genome, cccDNA quantification from rhesus macaque PH was normalized by digesting HBV DNA, extracted from HBV stocks and thus containing only rcDNA, with T5 exonuclease and measuring quantification of this product with both primer sets.

**hNTCP DNA and RNA quantification**. To quantify human NTCP DNA and RNA in macaque liver samples, total DNA and RNA were extracted using the AllPrep DNA/RNA kit according to manufacturer's instructions (Qiagen). The SuperScript VILO cDNA Synthesis Kit (Thermo Fischer) was used for cDNA synthesis according to manufacturer's instructions. Amplification of the macaque single copy gene PrP was used for normalization. Following primers were used: hNTCP_qPCR-F (5′-TGCCTCAATGTTCTTCAGCC-3′), hNTCP_qPCR-R (5′-TGTTCATGTTGTTCTTCATC-3′), mPrP_qPCR-F (5′-TGCTGGGAAGTGC-CATGAG-3′), and mPrP_qPCR-R (5′-CGGTACATGTTTTCACGATAGTA-3′) using an annealing temperature of 58 °C. All thermocycling and quantification measurements were conducted on an Applied Biosystems Step One PLUS using PerfeCTa SYBR Green FastMix, ROX (Quantabio) and a 3-step a PCR cycling protocol. Quantification was assessed on DNA level relative to macaque DNA and human DNA for mPrP and hNTCP, respectively. RNA level quantification was assessed relative to an absolute standard curve using plasmid DNA containing the entire hNTCP sequence.

**Southern blot analysis**. DNA was extracted from HBV-infected rhesus macaque PH using a modified Hirt extraction, as previously described[21]. Hirt DNA was separated on an 1.2% agarose gel and transferred onto a positively charged nylon membrane via upward capillary transfer. Hirt DNA extracted from HBV-infected HepG2-NTCP cells were loaded in parallel to show cccDNA appears at the same position[45]. To confirm the presence of cccDNA, Hirt extracted DNA was digested with T5 exonuclease (New England Biolabs) at 37 °C for 30 min. Linear HBV DNA fragments (3.2 kb, 2.2 kb, 1.9 kb and 1.4 kb) serve as a size marking ladder. The membrane was subjected to UV-crosslinking to covalently bind the transferred DNA onto it. The membrane was then hybridized with a digoxigenin-labeled HBV-specific probe, washed, and visualized by chemiluminescence according to manufacturer's instructions (DIG Luminescent Detection Kit).

**Analysis of NTCP**. For Fig. 2a, the topology of the human NTCP from Uniprot (http://www.uniprot.org/uniprot/Q14973) was used to generate a schematic illustration of the secondary structure of the protein. In addition, this amino acid sequence was compared to the macaque sequence (http://www.uniprot.org/uniprot/G7PAR4) using BLASTP 2.6.0 + (https://blast.ncbi.nlm.nih.gov/Blast.cgi). Differences in the sequences were labeled with lighter red for amino acid exchanges with similar physiochemical properties and darker red for exchanges with different physiochemical properties.

To determine hNTCP expression, rhesus macaque PH were collected from wells with 0.05% trypsin media (Gibco) and washed twice in HBSS (200×g for 5 min). Cells were incubated with MyrcludexB-Alexa488, which specifically binds human NTCP, for 30 minutes at room temperature, washed twice in HBSS, fixed with 2% paraformaldehyde (Electron Microscopy Sciences), and then collected on a Becton-Dickenson LSR-II. Analysis was performed on FlowJo X (TreeStar Inc.).

**ELISA**. Anti-HBs, Anti-HBe, HBsAg and HBeAg ELISA were performed on the Architect i1000 SRTM (Abbott) and anti-HBc ELISA on the BEP III (Siemens) per the manufacturer's instructions. In addition, HBsAg ELISA were performed using the Hepatitis B Surface Antigen BioAssay ELISA Kit (US Biological Life Sciences).

Anti-hNTCP ELISA were custom assays performed at the ONPRC. 96 well plates were coated with 0.25 µg per well recombinant hNTCP (Origene) in 50 µl overnight and then blocked with 150 µl Blotto (5% dry milk and 1% normal goat serum in PBS) for 1 h. Serum was heat inactivated at 56 °C for 30 min, serially diluted threefold, and 50 µl added to each well. Tests were performed in duplicate. Following serum incubation for 1 h at room temperature, plates were washed and

goat anti-human IgG-HRP (Jackson Labs) was added in 50 µl and incubated for 1 h at room temperature. Following secondary antibody incubation, plates were washed and TMB substrate (Southern Biotech) added in 50 µl and incubated for 15 min at room temperature. Volume of 50 µl per well of 1 N sulfuric acid was added to stop the reaction and plates were read on a SpectraMAX 190 (Molecular Devices) at 450 nm.

**IFNγ ELISpot.** Rhesus macaque liver-resident mononuclear cells were isolated from single-cell suspensions generated from collagenase media perfusion of livers, and peripheral blood mononuclear cells were isolated from blood. Pre-coated IFN-γ ELISpot^PLUS plates (Mabtech Inc.) were used to detect T cell responses and experiments were conducted per manufacturer's recommendations in triplicate wells as previously described[46]. Each well contained $1 \times 10^5$ cells in 100 µl complete T-cell media (RPMI 1640, 10% bovine growth serum (HyClone), supplemented with L-glutamine (HyClone) and antibiotic/antimycotic (HyClone)). Liver-resident T cells were incubated with 5 µM 15-mer peptide pools spanning the HBV open reading frames C, P, S, and X (70% peptide purity or greater). Peripheral blood mononuclear cells were incubated with 2 mM recombinant hNTCP protein. Wells were imaged with an AID ELISPOT reader and spots were counted by an auto-mated system with set parameters for size, intensity, and gradient. Responses were considered positive and statistically significant if the mean number of spot forming cells (SFCs) of triplicate sample wells exceed background plus two standard deviations. Responses less than 50 SFCs per million cells were considered negative (below the limit of detection).

**Immunohistochemistry.** Sections (2 µm) of 4% paraformaldehyde-fixed, paraffin-embedded liver samples were stained using a BondMAX immunohistochemistry robot (Leica) using Bond solutions and Polymer Refine Detection Kit (Leica). Primary antibody HBcAg (RB1413A, StartFragmentLab VisionEndFragment) was used at a 1:50 dilution. Primary antibody was detected with Bond Polymer Refine Detection Kit from Leica (rabbit, Leica, DS9800).

For quantification of HBcAg staining, whole slides were scanned at ×40 magnification using a SCN400 slide scanner (Leica) and analyzed withTissue IA image analysis software (Slidepath, Leica) using an optimized color detection setting for DAB (positive) and Hematoxilin (negative) nuclei at ×20 magnification. For quantification, 3 random areas of 5–10 mm$^2$ in size on each slide were chosen and obtained values were merged for further statistical analysis. For HBcAg-positive hepatocytes, data are presented as positive nuclei (DAB positive nuclei) as a percentage of total nuclei in liver (DAB/Hematoxylin+, including hepatocytes and non-parenchymal cells).

**Electron microscopy.** Supernatants from rhesus macaque PH, confirmed for infection with HBV (Fig. 2c), were diluted 1:10 in 4% paraformaldehyde. 10 µl of the fixed viral suspensions were deposited onto glow discharged carbon formvar 400 Mesh copper grids (Ted Pella 01822-F) for 3 min, rinsed 15 s in water, wicked on Whatman filter paper 1, stained for 60 s in filtered 1.33 % (w/v) uranyl acetate in water, wicked and air dried. EM grids were imaged at 120 kV on a FEI Tecnai Spirit TEM system. Images were acquired as 2048 × 2048 pixel, 16-bit gray scale files using the FEI's TEM Imaging and Analysis (TIA) interface on an Eagle 2 K CCD multiscan camera. All the images were acquired at 2–4 microns underfocused to improve sample contrast.

**Data availability.** All relevant data presented here are available from the authors without restrictions. The consensus sequence of the HBV stock used to infect rhesus macaques (HBV isolated from HepAD38) is available through GenBank Accession #MF967563. The mcHBV consensus sequence has already been pub-lished and can be accessed through GenBank Accession #HE815465.1.

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

## Acknowledgements

We thank Ke Zhang for cloning HBV1.3-P68S, Stephan Urban for providing Myrcludex B, Tanja Bauer for providing HBV peptide sets, Christian Lanciault for IHC analysis, and Mei Yu and Guofeng Cheng from Gilead Sciences, Inc for providing their HBV cccDNA qPCR protocol. Electron microscopy was performed at the Multiscale Microscopy Core (MMC) with technical support from the Oregon Health and Science University (OHSU)-FEI Living Lab and the OHSU Center for Spatial Systems Biomedicine (OCSSB). This study was funded in part by an ONPRC pilot project grant to J.B.S. and P51 OD011092 to ONPRC and by the German Research Foundation (DFG) via the Collaborative Research Center TRR179. We thank the animal care staff at ONPRC for their outstanding care of all study animals.

## Author contributions

B.J.B. wrote the manuscript. B.J.B., J.M.W., M.A.M.-H., C.K., M.M.F., K.E., T.Q.V., A.A., N.L.H., U.P. and J.B.S. designed the experiments. B.J.B., J.M.W., M.A.M.-H., M.R., C.K., M.M.F., K.B.H., M.N., B.N.B., T.J., J.S.R., R.N., S.M.-T., J.M.G., H.L.W., S.A., G.W. and W.F.S. conducted the experiments. B.J.B., J.M.W., M.A.M.-H., M.R., C.K., M.M.F., T.Q.V., A.A., N.L.H., U.P. and J.B.S. edited the manuscript. B.P. performed the statistics. A.K., T.S. and A.W.L. performed animal procedures.
