## [Peer Review File · Nature Communications]

Reviewers' comments:

Reviewer #1 (Remarks to the Author):

The manuscript by Burwitz and colleagues describes the development of a macaque model to study HBV infection. HBV research has been limited because of the lack of highly relevant animal models. Therefore, the results presented here represent a significant advance to the field that will be of great interest to the readers of Nature Communications. The data showing HBV replication by DNA detection is compelling and leaves no doubt that the HBV is replicating in some of the animals. Overall, the results presented are robust and support the conclusions of the authors. The only major issue is the IHC presented in figure 3E, which shows low resolution images and the staining seems to be a bit off.

1) Shouldn't be so quick to dismiss the transient Alt signal in last panel in figure 1. Animal 33451 has both transient ALT signal and CD3 increase. It would be great if the panels in figure 1 could have better labeling (E, F, G, etc). It a bit confusing as presented and will likely be very small in final publication.

2) The IHC images in figure 3E are very important data. But the resolution is poor to appreciate the details and the staining is not very compelling. It is important for the authors to provide better images that make a more compelling argument. Putting the negative controls first might help. A better presentation of this key data is necessary for the reader to be convinced that this data supports the conclusions made by the authors. However, there is no doubt that some of the animals are being infected with HBV based on the helper-dependent adenovirus vectors expressing NTCP treated animals shown in the panels in Fig 3.

3) It would be helpful if the authors took some time to discuss the differences observed in the animals that received the AAV8-hNTCP and HDAd-hNTCP. As a reader I found myself wondering about these differences and it would be useful for the authors to contribute their thoughts to the presented in vivo data which is fascinating.

Reviewer #2 (Remarks to the Author):

In this manuscript, the authors presented two findings. The first one regarded their failure in duplicating the important finding about one HBV strain, from Mauritian cynomolgus macaque, in natural infection. They could not infect this strain(mCHBV) or human HBV to other Mauritian cynomolgus macaques, even after depletion of CD3+ cells from these non-human primates. The second point regards the transduction of human HBV receptor, huNTCP, into hepatocytes from macaques or baboons, could render them susceptible to HBV infection in cell cultures. Moreover, they tried to transduce huNTCP into the liver of living macaques, and then inoculate them with HBV. After infection, they discovered some evidence of HBV infection and replication in their liver or blood.

Comments:

The development of a non-primate model for chronic HBV infection remains an unmet challenge in the research community. The efforts of HBV infection in tupaia model made some progress, but non-human primate is certainly a even better one. The first part was quite clear as they could not reproduce the infection of mCHBV strain in the Mauritian cynomolgus macaques. The information may be valuable to the reserachers in this field.

The second part, a transduction of huNTCP into the hepatocytes of macaque or baboons in vitro or in vivo, can convert these cells or the animals permissive to HBV infection, is an even more important progress. However, the data only appeared to be interesting, but not conclusive.

1. For the HBV infection of huNTCP-expressing non-primate hepatocytes, the level of supernatant HBV DNA, HBsAg, HBeAg was varying a lot between adult versus neonatal hepatocytee (about 40 fold-difference_. This needs a careful explanation (Fig. 2C vs. 2G). In addition, the southern blot for HBV cccDNA has to be collaborated with restraction enzyme digestion. In fact, a southern blot for total HBV DNA in the infected cells might show more viral replication intermediates.

2. The HBV infection of hu-NTCP trasduced macaques posed more problems. The authors had better showed the efficiency of huNTCP transduction in the liver of macaques. In addition, for the immuno-staining data, the photos showed very heavy background and difficult to read. More important, they need to do dobule-immuno-staining for huNTCP and HBV core or HBsAg colocalization, as only huNTCP expression hepatocytes presumably infected.

3. The serum HBV markers, including HBsAg, HBeAg, are better shown, in addition to a very low level HBV DNA. The dynamic of anti-HBc or anti-HBe, anti-HBs were poorly correlated. Finally, the intrahepatic HBV RNA levels appeared lower than that of HBV DNA, this is against the HBV biology and need more data (northern and southern blots)

4. Does the transduced huNTCP, as foreign protein, induce antibody or T cell immune reactions in macaques ? Please check this.

Reviewer #3 (Remarks to the Author):

This important study shows that following introduction of the NTCP receptor, rhesus macaques are susceptible to HBV infection, suggesting a new animal model of HBV infectivity and replication that may be applicable to testing new therapies. The authors acknowledge a major limitation of their model is the ability to introduce NTCP to large numbers of macaque hepatocytes.

Comments:

The cccDNA Southern blot on Rhesus PH is not convincing- this is a total DNA prep with a faint band of DNA that is similar in size to cccDNA extracted by HIRT digestion from NTCP-HepG2 cells. A HIRT extraction should have been performed and ExoI/III nuclease digestion performed on the DNA, as well as the T5 exonuclease. For the cccDNA PCR, what controls were used to rule out amplification of RC DNA?

Was cccDNA HIRT extraction and PCR/Southern blot performed on the animal derived liver tissue?

HBV RNA intermediates should be shown(Northern blots).

Did the authors measure anti-HBe in the animals?

Can the authors detect virus in the serum of infected macaques following IP and Southern blot? What is the profile of secreted particles (virions and subviral particles) compared to human HBV infection?

Fig. 3E is much too small to interpret.

Reviewer #1:

Summary: The manuscript by Burwitz and colleagues describes the development of a macaque model to study HBV infection. HBV research has been limited because of the lack of highly relevant animal models. Therefore, the results presented here represent a significant advance to the field that will be of great interest to the readers of Nature Communications. The data showing HBV replication by DNA detection is compelling and leaves no doubt that the HBV is replicating in some of the animals. Overall, the results presented are robust and support the conclusions of the authors. The only major issue is the IHC presented in figure 3E, which shows low resolution images and the staining seems to be a bit off.

**Author Response: We thank reviewer 1 for these positive comments on our work. As described below, we have optimized the IHC staining which now clearly shows the presence of HBc in macaque hepatocytes following hNTCP expression and HBV challenge.*

Reviewer #1 Comment 1: Shouldn't be so quick to dismiss the transient Alt signal in last panel in figure 1. Animal 33451 has both transient ALT signal and CD3 increase. It would be great if the panels in figure 1 could have better labeling (E, F, G, etc). It a bit confusing as presented and will likely be very small in final publication.

**Author Response: We agree with the reviewer that the concomitant increase of ALT and CD3 absolute count should not be immediately dismissed. We have adjusted the text to point out the parallel ALT/CD3 increase, but still make the final point that no signs of HBV infection were present. In addition, we have reconfigured Figure 1 to include more panel labels, as suggested by the reviewer.*

Reviewer #1 Comment 2: The IHC images in figure 3E are very important data. But the resolution is poor to appreciate the details and the staining is not very

compelling. It is important for the authors to provide better images that make a more compelling argument. Putting the negative controls first might help. A better presentation of this key data is necessary for the reader to be convinced that this data supports the conclusions made by the authors. However, there is no doubt that some of the animals are being infected with HBV based on the helper-dependent adenovirus vectors expressing NTCP treated animals shown in the panels in Fig 3.

Author Response: We agree with Reviewer 1 that the IHC in Figure 3E is important data and should be presented more clearly. To address this point, we tested multiple HBc antibodies and optimized the staining protocol, which is newly described in the methods section on line 445. Additionally, we also now include high resolution images to more clearly show the presence of HBc in hepatocytes. These data are now presented in a revised Figure 3E.

Reviewer #1 Comment 3: It would be helpful if the authors took some time to discuss the differences observed in the animals that received the AAV8-hNTCP and HDAd-hNTCP. As a reader I found myself wondering about these differences and it would be useful for the authors to contribute their thoughts to the presented in vivo data which is fascinating.

**Author Response: We fully agree that more discussion of the differences between AAV and HDAd vectors warrants further discussion. We now outline the differences in the vectors and discuss their impact on line 225.*

Reviewer #2:

Summary: In this manuscript, the authors presented two findings. The first one regarded their failure in duplicating the important finding about one HBV strain, from Mauritian cynomolgus macaque, in natural infection. They could not infect this strain (mcHBV) or human HBV to other Mauritian cynomolgus macaques, even after depletion of CD3+ cells from these non-human primates. The second point regards the transduction of human HBV receptor, huNTCP, into hepatocytes from macaques or baboons, could render them susceptible to HBV infection in cell cultures. Moreover, they tried to transduce huNTCP into the liver of living macaques, and then inoculate them with HBV. After infection, they discovered some evidence of HBV infection and replication in their liver or blood.

The development of a non-primate model for chronic HBV infection remains an unmet challenge in the research community. The efforts of HBV infection in tupaia model made some progress, but non-human primate is certainly an even better one. The first part was quite clear as they could not reproduce the infection of mcHBV strain in the Mauritian cynomolgus macaques. The information may be valuable to the researchers in this field.

The second part, a transduction of huNTCP into the hepatocytes of macaque or baboons in vitro or in vivo, can convert these cells or the animals permissive to HBV infection, is an even more important progress. However, the data only appeared to be interesting, but not conclusive.

**Author Response: We thank reviewer 2 for their perspective on this work. We believe the data presented, particularly the longitudinal HBV viral loads present in serum, conclusively show that HBV can replicate in rhesus macaques in vivo following transduction of hepatocytes with human NTCP. However, we agree with the reviewer that further evidence to localize this replication to the liver would be*

beneficial. To this end, we have analyzed additional samples from the rhesus macaque livers and optimized our HBc IHC protocol to definitively show that HBV gene expression and replication are found in the liver.

Reviewer #2 Comment 1: For the HBV infection of huNTCP-expressing non-primate hepatocytes, the level of supernatant HBV DNA, HBsAg, HBeAg was varying a lot between adult versus neonatal hepatocytes (about 40 fold-difference). This needs a careful explanation (Fig. 2C vs. 2G). In addition, the southern blot for HBV cccDNA has to be collaborated with restriction enzyme digestion. In fact, a southern blot for total HBV DNA in the infected cells might show more viral replication intermediates.

**Author Response: The purpose of the infection of neonatal hepatocytes was to determine the persistence of infection over time in macaque hepatocytes in vitro. A comparison to adult hepatocytes is not intended. Thus, we only show HBsAg levels secreted from HBV-infected neonate primary hepatocytes (Fig. 2G) in a qualitative and not in a quantitative fashion. The experiment shown in Fig. 2C was meant to directly compare AAV- with HDAd-vectors expressing hNTCP, and therefore a quantitative read-out was utilized. Because of this, the units presented along the y-axis are not comparable between these two panels. Overall, we believe that our major finding that HBV replicates in primary macaque hepatocytes expressing hNTCP is well supported by this data.*

We agree with the reviewer that further confirmation of cccDNA formation is of paramount importance, given the power of this model will depend on recapitulating all steps of the HBV life cycle. We now show direct comparison of Hirt extractions between rhesus macaque PH and HepG2-NTCP cells in Figure 2F. Importantly, cccDNA was resistant to T5 exonuclease digestion while the other HBV replication intermediates were readily digested.

Reviewer #2 Comment 2: The HBV infection of hu-NTCP transduced macaques posed more problems. The authors had better showed the efficiency of huNTCP transduction in the liver of macaques. In addition, for the immuno-staining data, the photos showed very heavy background and difficult to read. More important, they need to do double-immuno-staining for huNTCP and HBV core or HBsAg colocalization, as only huNTCP expression hepatocytes presumably infected.

**Author Response: We thank the reviewer for these suggestions, and now present qPCR results showing AAV8- or HDAd-dependent hNTCP transduction of liver in all four HBV infected rhesus macaques. Importantly, transduction is present and uniform across all 3 liver lobes assessed in each animal. Negative controls, which consisted of matched liver samples from untreated macaques, show no amplification of hNTCP DNA. These results are now included as supplemental figure 3. Additionally, as discussed above in response to reviewer 1, we optimized the HBcAg IHC staining protocol for macaque livers, which resulted in more specific staining of HBc with limited background staining. These data more definitively show positive staining and are now presented in a revised Figure 3E. Despite multiple attempts to co-stain our formalin-fixed, paraffin embedded liver tissues for hNTCP and HBcAg, we were not able to achieve staining for hNTCP using MyrB or commercially available antibodies. We believe this is most likely due to loss of the relevant epitopes during the formalin-based fixation process, as has been described previously for multiple antigens in IHC. We now discuss this point in the discussion on line 232. Nevertheless, our in vitro and in vivo results presented here definitively demonstrate that hNTCP is required for entry and replication of HBV in nonhuman primate hepatocytes. This*

finding is also in line with decades of failed attempts to infect untreated, non-hNTCP expressing macaques with HBV.

Reviewer #2 Comment 3: The serum HBV markers, including HBsAg, HBeAg, are better shown, in addition to a very low level HBV DNA. The dynamic of anti-HBc or anti-HBe, anti-HBs were poorly correlated. Finally, the intrahepatic HBV RNA levels appeared lower than that of HBV DNA, this is against the HBV biology and need more data (northern and southern blots).

Author Response: We now include measures of HBsAg, anti-HBe, and anti-HBs antibodies, collected at the time of euthanasia, in supplemental table 4. However, in line with the low HBV DNA levels observed, these parameters were all undetectable. Nevertheless, the lack of these markers do not dispute the fact that hNTCP facilitates HBV infection of macaques. Instead, it indicates that higher level of hNTCP will be necessary to support higher levels of HBV replication.

Unfortunately, the low levels of in vivo replication preclude northern and southern blots due to a paucity of nucleic acid mass available. However, in response to reviewers, we have repeated our in vitro southern blot utilizing Hirt-extracted DNA and we now show rcDNA and dlDNA, indicating that additional replicative forms of HBV are present. While we observed that HBV RNA levels were lower than DNA levels in vivo, we do not currently know if HBV RNA transcription differs from the human situation, potentially due to distinct hepatocyte-specific nuclear (transcription) factors which may be expressed at different levels in macaque hepatocytes than in human hepatocytes. While determining the transcription levels is an important next step, it is beyond the scope of the current study, which demonstrates the ability to utilize nonhuman primates for HBV research via expression of hNTCP. For the purposes of this publication, we believe the salient point is that RNA is detectable, indicating active replication in macaques, and that this study represents a significant milestone for the development of a nonhuman primate model of HBV infection.

Reviewer #2 Comment 4: Does the transduced huNTCP, as foreign protein, induce antibody or T cell immune reactions in macaques ? Please check this.

Author Response: We thank the reviewer for this suggestion, and agree that hNTCP specific immunity could affect the duration of HBV infection in macaques. We have measured both humoral and cellular immunity against hNTCP and found no antibody or T cell immune responses against the transgene in our HBV-infected rhesus macaques. This data is now shown in supplemental figure 4.

Reviewer #3:

Summary: This important study shows that following introduction of the NTCP receptor, rhesus macaques are susceptible to HBV infection, suggesting a new animal model of HBV infectivity and replication that may be applicable to testing new therapies. The authors acknowledge a major limitation of their model is the ability to introduce NTCP to large numbers of macaque hepatocytes.

Reviewer #3 Comment 1: The cccDNA Southern blot on Rhesus PH is not convincing- this is a total DNA prep with a faint band of DNA that is similar in size to cccDNA extracted by HIRT digestion from NTCP-HepG2 cells. A HIRT extraction should have been performed and ExoI/III nuclease digestion performed on the DNA, as well as the T5 exonuclease. For the cccDNA PCR, what controls were used to rule out amplification of RC DNA?

Author Response: Please see our response to reviewer #2 comment 1. We agree with the reviewer that further confirmation of cccDNA formation is of paramount importance, given the power of this model will depend on recapitulating all steps of the HBV life cycle. We now show direct comparison of Hirt extractions between rhesus macaque PH and HepG2-NTCP cells in Figure 2F. In addition, we show that HBV rcDNA and dlDNA are readily digested by T5 exonuclease, but cccDNA remains intact. This demonstrates that cccDNA is formed in rhesus macaque PH infected with HBV.

For the cccDNA qPCR shown in Figure 2E, which supports our southern blot analysis, we have updated our materials and methods on line 377 to more clearly articulate how cccDNA was quantified and normalized to account for potential amplification of T5 exonuclease digested rcDNA. As requested by the reviewer, this section now states: "To control for low-level amplification of rcDNA with the primer/probe set targeting the nick and gap of the HBV genome, cccDNA quantification from rhesus macaque PH was normalized by digesting HBV DNA, extracted from HBV stocks and thus containing only rcDNA, with T5 exonuclease and measuring quantification of this product with both primer sets."

Reviewer #3 Comment 2: Was cccDNA HIRT extraction and PCR/Southern blot performed on the animal derived liver tissue?

Author Response: Unfortunately, the low level of HBV replication in these rhesus macaques makes detection of HBV cccDNA in vivo difficult. Indeed, the levels of total HBV DNA ranged from 800 – 11,000 copies per 1 million perfused hepatocytes, making detection of cccDNA exceedingly difficult. However, we believe our in vitro data convincingly shows that cccDNA is formed in HBV-infected rhesus macaque PH.

Reviewer #3 Comment 3: HBV RNA intermediates should be shown (Northern blots).

Author Response: Unfortunately, the low level of HBV replication in these rhesus macaques makes detection of HBV RNA intermediates in vivo difficult. Indeed, the levels of total HBV RNA ranged from 30 – 70 copies per 1 million perfused hepatocytes, making detection of RNA intermediates exceedingly difficult. Therefore, we are unable to provide this data at this time. While we agree that northern blots demonstrating the RNA intermediates is a logical next step, we believe it is beyond the scope of the current study, which demonstrates the ability to utilize nonhuman primates for pre-clinical HBV research. Indeed, the presence of rcDNA, dlDNA, cccDNA and Dane particles demonstrate that the entire HBV replication cycle is occurring in macaque hepatocytes expressing hNTCP.

Reviewer #3 Comment 4: Did the authors measure anti-HBe in the animals?

Author Response: Yes, we now include anti-HBe measurements from the time of euthanasia in supplemental table 4. We did not detect anti-HBe responses in these animals. This is likely a result of the low level in vivo infection, which indicates that higher expression of hNTCP in hepatocytes will be necessary for measurement of anti-HBe.

Reviewer #3 Comment 5: Can the authors detect virus in the serum of infected

macaques following IP and Southern blot? What is the profile of secreted particles (virions and subviral particles) compared to human HBV infection?

Author Response: Unfortunately, the volume of serum available from the final time points coupled with the low viral loads makes IP of sufficient virions technically impractical. However, we believe our in vitro data convincingly shows that Dane particles and HBsAg virion-like particles are released from HBV-infected rhesus macaque PH.

Reviewer #3 Comment 6: Fig. 3E is much too small to interpret.

Author Response: We agree with reviewer 3 and have optimized the staining parameters more thoroughly to obtain more definitive results (see response above to reviewers 1 and 2). Furthermore, we have increased the resolution and size of figure 3E as requested.

Reviewers' comments:

Reviewer #1 (Remarks to the Author):

The authors have done an excellent job of addressing the critiques of the initial review.

Reviewer #2 (Remarks to the Author):

The authors conducted several experiments to address the reviewers' comments. Basically, they presented a better southern blot to detect HBV cccDNA in the Macaques liver of infected liver transduced with hNTCP. They also presented a more clear IHC staining for HBV core antigen in that liver. The manuscript appeared to be improved.

Nonetheless, the reviewer remained to be convinced.

There are three points not so ensuring.

1. As it is well-known, the HBV RNA in the infected liver is 10 or 20 times more than viral DNAs. It is difficult to understand why the authors could not do a northern blot to detect viral RNAs, despite their success in using southern blot to show viral DNAs.
2. The immunostaining for HBV core antigen did find a few cells (what the percentages?). However, the signal is too few and needs other collaborating evidences, for example, the co-staining of HBsAg. Or more critical, the co-localization with hNTCP. Both data are missing.
3. It is intriguing about the authors' capability in showing T cell responses to HBV antigens in infected Macaques, but not able to demonstrate the B cell responses, such as anti-HBc, which are much more common to detect in naturally HBV infection systems.
4. To make a more conclusive result, either the authors need to increase the efficiency of hNTCP-transduction efficiency, or to apply immuno-suppressing agents to enhance HBV replications. Current data only provided a minimal level of evidences.

Reviewer #3 (Remarks to the Author):

The authors have adequately addressed the Reviewer's comments. The figures are much improved and limitations of the study are acknowledged.

The manuscript needs to be checked closely for spelling errors, ie antibodies is spelt incorrectly on line 232.

Response to reviewers' comments:

Reviewer #1:

Summary: The authors have done an excellent job of addressing the critiques of the initial review.

Author response: We thank the reviewer for his/her initial comments, which we believed strengthened the manuscript.

Reviewer #2:

Summary: The authors conducted several experiments to address the reviewers' comments. Basically, they presented a better southern blot to detect HBV cccDNA in the Macaques liver of infected liver transduced with hNTCP. They also presented a clearer IHC staining for HBV core antigen in that liver. The manuscript appeared to be improved.

Nonetheless, the reviewer remained to be convinced.

There are three points not so ensuring.

1. As it is well-known, the HBV RNA in the infected liver is 10 or 20 times more than viral DNAs. It is difficult to understand why the authors could not do a northern blot to detect viral RNAs, despite their success in using southern blot to show viral DNAs.

Author response: We fully intend to continue studying and optimizing this novel macaque model of HBV infection, and virology-focused follow-up studies to definitively characterize the replication of HBV in macaques are planned. However, we believe that the data we present here convincingly shows that macaques are susceptible to HBV infection, and that HBV actively replicates in vivo following expression of hNTCP in hepatocytes. There is little doubt that HBV transcripts are expressed, as viral particles can be monitored over time in the serum and HBc is expressed in the nuclei of infected cells. We believe that investigating the ratios of HBV DNA:RNA in the livers of HBV infected macaques would complicate the manuscript, and is therefore outside the scope of the current study. A subsequent study focused only on the replication cycle of HBV in macaque hepatocytes in comparison to human hepatocytes would be more detailed and informative, and would be geared towards readers of virology-specific journals.

2. The immunostaining for HBV core antigen did find a few cells (what are the percentages?). However, the signal is too few and needs other collaborating evidences, for example, the co-staining of HBsAg. Or more critical, the co-localization with hNTCP. Both data are missing.

Author response: We now present data in Figure 3 showing the overall frequency of HBcAg-positive nuclei in the livers of rhesus macaques. We also present data in Figure S1 showing DNA/RNA levels of hNTCP in the livers of AAV8 and HDAd-treated animals by qPCR/qRT-PCR, respectively. As mentioned in the discussion, previous work with AAV8-GFP vectors in mice and rhesus macaques indicates that hNTCP DNA quantities similar to ours ($10^2 - 10^0$ hNTCP DNA copies/diploid macaque genome) corresponds to less than 5% of hepatocytes expressing AAV delivered transgene. Therefore, the frequency of HBcAg-positive cells we observe in the liver is directly in line with the expected frequency of cells expressing hNTCP.

It is well documented that the human hepatoma cell line HepG2 requires hNTCP expression for robust infection, and that rhesus macaques (and their hepatocytes) cannot be naturally infected

by HBV. As we have shown *in vitro* and *in vivo*, exogenous expression of hNTCP is the definitive restriction to HBV infection of rhesus macaques (similar to HepG2). This finding is fully supported by recently published work by Stephan Urban's group, showing co-staining of primary macaque hepatocytes *in vitro* with MyrB and HBcAg (Lempp et al., *Hepatology*, 2017).

3. It is intriguing about the authors capability in showing T cell responses to HBV antigens in infected Macaques, but not able to demonstrate the B cell responses, such as anti-HBc, which are much more common to detect in naturally HBV infection systems.

Author response: As shown in Figures 3B and 3C, the animal with a statistically significant T cell response against HBsAg also exhibited anti-HBcAg seroconversion.

4. To make a more conclusive result, either the authors need to increase the efficiency of hNTCP-transduction efficiency, or to apply immuno-suppressing agents to enhance HBV replications. Current data only provided a minimal level of evidences.

Author response: We disagree with the reviewer and believe that the current data demonstrate, with little doubt, that HBV replicates in macaques and that anti-viral cellular and humoral immune responses are elicited. In addition, we clearly state in the discussion that the largest limitation to the model is low transduction rates in the liver, as also stated by the reviewer here. Increasing liver transduction should lead to a more robust model, and we are currently testing various means to optimize hNTCP expression on hepatocytes *in vivo*, including treatment with immunosuppressive agents.

Reviewer #3:

Summary: The authors have adequately addressed the Reviewer's comments. The figures are much improved and limitations of the study are acknowledged. The manuscript needs to be checked closely for spelling errors, ie antibodies is spelt incorrectly on line 232.

Author response: We thank the reviewer for his/her comment, and have closely reviewed the manuscript in an attempt to identify and rectify all misspellings.

REVIEWERS' COMMENTS:

Reviewer #2 (Remarks to the Author):

The authors responded to the four questions only by arguments, believing their current data convincing enough to support a HBV infection in hNTCP-transduced macaques. Maybe a little bit more experimental results are still needed.